# Cost-effectiveness of COVID rapid diagnostic tests for patients with severe/critical illness in low- and middle-income countries: A modeling study

Gabrielle Bonnet[1]*, John Bimba[2,3], Chancy Chavula[4], Harunavamwe N. Chifamba[5], Titus H. Divala[6], Andres G. Lescano[7], Mohammed Majam[8], Danjuma Mbo[9], Auliya A. Suwantika[10,11], Marco A. Tovar[12,13], Pragya Yadav[14], Obinna Ekwunife[15,16], Collin Mangenah[17], Lucky G. Ngwira[18], Elizabeth L. Corbett[19], Mark Jit[1‡], Anna Vassall[20,21‡]

1 Department of Infectious Disease Epidemiology, London School for Hygiene and Tropical Medicine, Faculty of Public Health and Policy, London, United Kingdom, 2 Zankli Research Centre, Bingham University, Karu, Nigeria, 3 Department of Community Medicine, Bingham University, Karu, Nigeria, 4 Clinton Health Access Initiative, Lilongwe, Malawi, 5 Sally Mugabe Central Hospital, Harare, Zimbabwe, 6 Kamuzu University of Health Sciences (KUHeS), Blantyre, Malawi, 7 Emerge, Emerging Diseases and Climate Change Research Unit, School of Public Health and Administration, Universidad Peruana Cayetano Heredia, Lima, Peru, 8 Ezintsha, Faculty of Health Sciences, University of the Witwatersrand, Johannesburg, South Africa, 9 Maitama Hospital, Abuja, Nigeria, 10 Department of Pharmacology and Clinical Pharmacy, Faculty of Pharmacy, Universitas Padjadjaran, Bandung, Indonesia, 11 Center of Excellence for Pharmaceutical Care Innovation (PHARCI), Universitas Padjadjaran, Bandung, Indonesia, 12 Socios En Salud Sucursal Perú, Lima, Peru, 13 Escuela de Medicina, Universidad Peruana de Ciencias Aplicadas, Lima, Perú, 14 Indian Council of Medical Research National Institute of Virology, Pune, India, 15 Department of Clinical Pharmacy and Pharmacy Management, Nnamdi Azikiwe University, Awka, Nigeria, 16 Department of Medicine, University at Buffalo, Buffalo, New York, United States of America, 17 Centre for Sexual Health, HIV and AIDS Research, Harare, Zimbabwe, 18 Health Economics Policy Unit, Kamuzu University of Health Sciences, Blantyre, Malawi, 19 Department of Clinical Research, London School of Hygiene & Tropical Medicine, Faculty of Public Health and Policy, London, United Kingdom, 20 Department of Global Health and Development, London School of Hygiene & Tropical Medicine, London, United Kingdom, 21 Amsterdam Institute for Global Health and Development, Amsterdam, the Netherlands

‡ These authors are joint senior authors on this work.
* gabrielle.bonnet@lshtm.ac.uk

**Data Availability Statement:** This paper uses only published/publicly available data, which are

## Abstract

### Background

Rapid diagnostic tests (RDTs) for coronavirus disease (COVID) are used in low- and middle-income countries (LMICs) to inform treatment decisions. However, to date, it is unclear when this use is cost-effective. Existing analyses are limited to a narrow set of countries and uses. The aim of this study is to assess the cost-effectiveness of COVID RDTs to inform the treatment of patients with severe illness in LMICs, considering real world practice.

### Methods and findings

We assessed the cost-effectiveness of COVID testing across LMICs using a decision tree model, differentiating results by country income level, Severe Acute Respiratory Syndrome Coronavirus 2 (SARS-CoV-2) prevalence, and testing scenario (none, RDTs, polymerase

described in this paper and supplements. No primary data were collected. The model code is available at: https://github.com/gbn0931/tcov.

**Funding:** GB, ELC, MJ and AV were supported by UNITAID/PSI grant 100584IR. AGL was sponsored by Emerge, the Emerging Diseases Epidemiology Research Training grant D43 TW007393 awarded by the Fogarty International Center of the US National Institutes of Health. The funders had no role in study design, data collection and analysis, decision to publish, or preparation of the manuscript.

**Competing interests:** The authors have declared that no competing interests exist.

**Abbreviations:** COVID, coronavirus disease; DALY, disability-adjusted life year; GBD, Global Burden of Disease; ICU, intensive care unit; LMIC, low- and middle-income country; LRTI, lower respiratory tract infection; MV, mechanical ventilation; PCR, polymerase chain reaction; RDT, rapid diagnostic test; SARS-CoV-2, Severe Acute Respiratory Syndrome Coronavirus 2; TB, tuberculosis; TCZ, tocilizumab; URTI, upper respiratory tract infection; YLL, year of life lost.

chain reaction tests—PCRs and combinations). LMIC experts defined realistic care pathways and treatment options. Using a healthcare provider perspective and net monetary benefit approach, we assessed both intended (COVID symptom alleviation) and unintended (treatment side effects) health and economic impacts for each testing scenario. We included the side effects of corticosteroids, which are often the only available treatment for COVID. Because side effects depend both on the treatment and the patient's underlying illness (COVID or COVID-like illnesses, such as influenza), we considered the prevalence of COVID-like illnesses in our analyses.

We found that SARS-CoV-2 testing of patients with severe COVID-like illness can be cost-effective in all LMICs, though only in some circumstances. High influenza prevalence among suspected COVID cases improves cost-effectiveness, since incorrectly provided corticosteroids may worsen influenza outcomes. In low- and some lower-middle-income countries, only patients with a high index of suspicion for COVID should be tested with RDTs, while other patients should be presumed to not have COVID. In some lower-middle-income and upper-middle-income countries, suspected severe COVID cases should almost always be tested. Further, in these settings, negative test results in patients with a high initial index of suspicion should be confirmed through PCR and, during influenza outbreaks, positive results in patients with a low initial index of suspicion should also be confirmed with a PCR. The use of interleukin-6 receptor blockers, when supported by testing, may also be cost-effective in higher-income LMICs. The cost at which they would be cost-effective in low-income countries ($162 to $406 per treatment course) is below current prices.

The primary limitation of our analysis is substantial uncertainty around some of the parameters in our model due to limited data, most notably on current COVID mortality with standard of care, and insufficient evidence on the impact of corticosteroids on patients with severe influenza.

## Conclusions

COVID testing can be cost-effective to inform treatment of LMIC patients with severe COVID-like disease. The optimal algorithm is driven by country income level and health budgets, the level of suspicion that the patient may have COVID, and influenza prevalence. Further research to better characterize the unintended effects of corticosteroids, particularly on influenza cases, could improve decision making around the treatment of those with COVID-like symptoms in LMICs.

## Author summary

### Why was this study done?

- The main role of Severe Acute Respiratory Syndrome Coronavirus 2 (SARS-CoV-2) testing has evolved from transmission reduction to informing the treatment of patients with most vulnerability or most severe illness.

- SARS-CoV-2 testing availability and use remains inconsistent in some low- and middle-income countries (LMICs), and understanding the cost-effectiveness of testing in such contexts may help guide priority setting and inform guidelines for health professionals.

- Research on the cost-effectiveness of SARS-CoV-2 testing in LMICs is limited. To our knowledge, no paper has considered both presumptive and symptomatic coronavirus disease (COVID) treatment as alternatives to testing, or has considered a range of treatment options reflective of LMIC contexts.

## What did the researchers do and find?

- We used a decision tree model, health payer provider perspective and net monetary benefit approach to assess the cost-effectiveness of testing to support treatment for patients with severe/critical COVID-like illness in 129 LMICs, based on treatment pathways reported by experts working in LMICs.

- In low-income and the poorest of lower-middle-income countries, only patients with a high index of suspicion for COVID should be tested with rapid diagnostic tests.

- In the wealthiest among lower-middle-income countries and in upper-middle-income countries, testing of suspected severe COVID cases is almost always recommended. Polymerase chain reaction (PCR) confirmation of negative test results in patients with a high index of suspicion and, during influenza epidemics/outbreaks, of positive test results in patients with a low index of suspicion, is recommended.

## What do these findings mean?

- COVID testing of patients with severe, COVID-like symptoms in LMICs can be cost-effective, provided sufficiently specific clinical screening algorithms can be used.

- Policymakers should consider both our study's results regarding variability between countries at a similar income level and its sensitivity analysis, particularly regarding country-specific factors, alongside other considerations such as local feasibility and equity, to inform national level decision-making.

- The main limitation of the study is uncertainty on some parameters, including current COVID case fatality risks and the impact of corticosteroids on patients with influenza.

## Introduction

The role of Severe Acute Respiratory Syndrome Coronavirus 2 (SARS-CoV-2) testing has evolved over the course of the coronavirus disease (COVID) pandemic. Initially, testing was used to support transmission reduction, enable release from quarantine/isolation, target treatment for infected patients, and support surveillance. With vaccination available and countries largely relaxing physical distancing practices [1,2], the importance of incidence reduction measures has diminished. However, the impact of COVID on human health remains substantial

and testing as part of treatment pathways remains potentially important, provided it leads to better care [3]. Testing is still recommended for the most vulnerable patients and those with severe illness, and testing information, advice, and guidance in low and middle-income countries (LMICs) such as India [4], Nigeria [5], Peru [6], and South Africa [7] typically links testing to isolation, treatment/care, or variant detection.

However, testing availability and use remains inconsistent across countries [8]. Understanding the impact and value of SARS-CoV-2 testing may help guide priority setting and inform guidelines for health professionals. When targeted at mild low-risk patients, testing potentially reduces inappropriate antibiotic use [8]. Patients at high risk for severe disease may receive oral antivirals, and patients with severe or critical illness presenting to care centers may receive corticosteroids, interleukin-6 (IL-6) receptor blockers, and/or other recommended COVID therapeutics [9]. To date, however, there are only a limited number of economic evaluations of COVID testing in LMICs, and even fewer consider antigen-based rapid diagnostic tests (RDTs) [10,11] as part of treatment pathways. To our knowledge, no previous study has considered both presumptive and symptomatic COVID treatment, estimated the unintended effects of treating false positives, or assessed a similar range of treatment options (e.g., corticosteroids and/or IL-6 receptor blockers) (see Section A in S1 Appendix for further details).

Here, we concentrate on the use of tests for patients with severe and critical illness presenting with symptoms associated with COVID ("suspected cases") at health facilities in LMICs. We did not model community use or in those with mild symptoms, due to the lack of knowledge of the impact of testing on health-seeking behavior. We assess the cost-effectiveness of COVID RDTs across a wide range of countries and SARS-CoV-2 prevalence, considering health impacts comprehensively. Our study and its focus were informed by an expert consultation [8] on COVID testing and clinical practices in LMICs to ensure local viewpoints and experience were included in our analysis.

## Method

We assessed the cost-effectiveness of RDT use for patients with severe COVID in 129 LMICs by estimating the cost per disability-adjusted life year (DALY) averted over a lifetime horizon. We compared the use of RDTs to a no-testing scenario, in contexts with/without polymerase chain reaction tests (PCRs) and with different confirmatory testing practices. We applied a health provider perspective. Study methods are reported as per the Consolidated Health Economic Evaluation Reporting Standards 2022 (CHEERS 2022) checklist (Section B in S1 Appendix).

### Model overview

We used a decision tree model to estimate the cost-effectiveness of RDTs to support targeted treatment for severe/critical suspected cases presenting at health facilities and requiring some form of oxygen supplementation. "Severe/critical illness" were defined in accordance with the WHO therapeutic guidelines [9] (at least one of: oxygen saturation <90% on room air, signs of pneumonia, severe or acute respiratory distress, sepsis, septic shock, or conditions needing life-sustaining therapies). The model was applied to 129 LMICs (list in Section B in S2 Appendix, with corresponding country-specific parameter values), using the 2022 World Bank classification and excluding countries with fewer than 90,000 inhabitants or with insufficient data (North Korea, Kosovo). Due to data limitations, results do not reflect individual country contexts sufficiently hence are not meant to be applied in individual countries without consideration of contextual factors.

## Country context: Results of an experts' consultation

The development of the decision tree was informed by a consultation with LMIC experts from India, Indonesia, Malawi, Nigeria, Peru, South Africa, and Zimbabwe (results published elsewhere [8]). This consultation explored common COVID screening, testing and management practices, availability of therapeutics, and linkage to care. The consultation highlighted that a positive test in a mild (low- or high-risk) patient may lead, depending on the country, to either increased or decreased linkage to care upon clinical deterioration. Further, out of the many COVID therapeutics recommended in WHO guidelines [9], only 2, corticosteroids and tocilizumab (TCZ), an IL-6 receptor blocker, are likely to be available for patients with severe illness, with the latter only reported available in higher-resource settings. The availability of conventional oxygen ($O_2$) and mechanical ventilation (MV) can also be limited, particularly MV. Finally, PCR tests are often, but not always, available as an alternative or complement to RDTs.

Experts and country guidance suggested that the choice to provide respiratory support primarily depends on the patients' clinical presentation; hence, the "real world" benefit of testing lies in its ability to inform the use of corticosteroids and/or TCZ. As these treatments are recommended only for patients needing oxygen supplementation [9], this narrowed our focus to patients with severe or critical illness needing $O_2$ or MV.

## Treatment scenarios

Given the resource constraints in some LMICs (as reported by the country experts), we explore 2 treatment scenarios. Treatment scenario 1 is where TCZ is unavailable and corticosteroid treatment is the only option. Treatment scenario 2 is where TCZ is available. Where TCZ treatment is available we compare 2a, a scenario where all positive patients may receive it alongside corticosteroids, and 2b, where positive patients only receive corticosteroids. This comparison helps assess whether the addition of TCZ is more cost-effective than simple corticosteroid use.

For the comparators with no testing, we assume untested suspect cases may either be (i) treated as not having COVID (receiving $O_2$/MV but not corticosteroids/TCZ), in which case incoming patients would still undergo clinical assessment for other diseases but not for COVID; or (ii) treated as having COVID (receiving corticosteroids and/or TCZ presumptively as in the scenarios above), in which case clinical assessment of COVID would be needed. Finally, we assumed that antibiotics are sometimes given to patients with severe illness, for example, to address secondary bacterial infections with their level of use varying based on test results [8].

## Testing algorithms

We compared 5 testing options for patients that are suspected of having COVID based on clinical assessment: no testing, using RDTs alone, using PCRs alone, or using RDTs and confirming test-negatives (or, in an alternative scenario, test-positives) with PCR. We did not factor in differences in treatment delay between testing pathways as experts reported that PCR results are now generally available within 24 h [8].

## Decision tree

The decision tree (**Fig 1**) represents all diagnostic and treatment pathways described above. When "recovery" is the end-point of a branch, the patient recovers from acute COVID but may nevertheless suffer from treatment side effects or post-acute Coronavirus Syndrome [12],

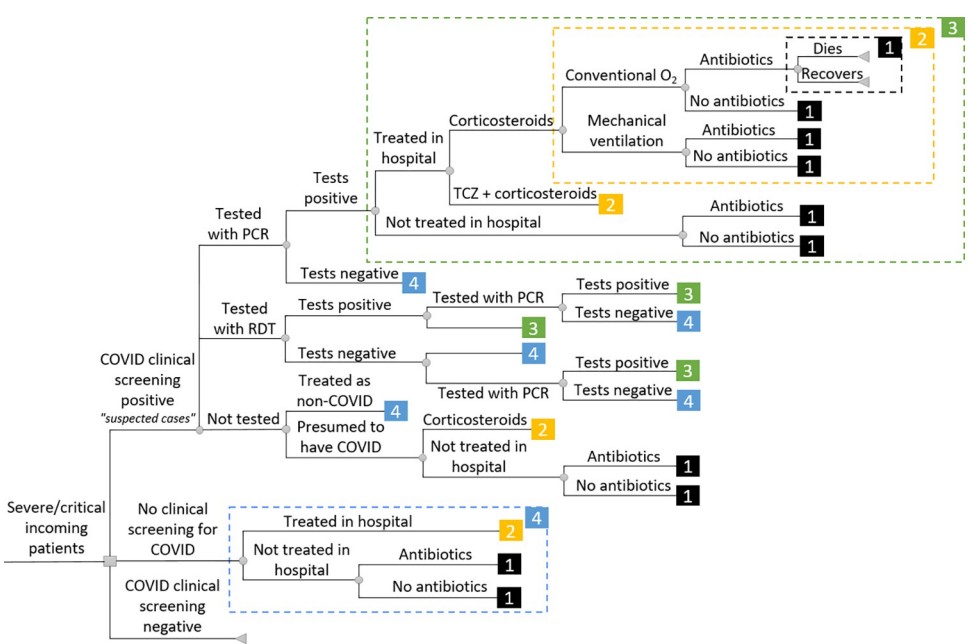

**Fig 1. Decision tree**\*. \* Each number corresponds to a segment of the decision tree (shown within the corresponding box with dashed outlines) that is repeated every time the number appears. For example, (1) corresponds to the death and recovery branches, (2) includes multiple options in terms of respiratory support (conventional oxygen and MV) and antibiotics (present or absent) and their consequences (dying or recovering), (3) combines patients not treated in hospital despite their severe or critical state and patients treated with either corticosteroids combined with the options delineated in (2), or tocilizumab and corticosteroids combined with (2), (4) includes scenarios that may be chosen in the absence of clinical screening for COVID (in this case, patients may only be treated for their symptoms and not as COVID patients, hence may only benefit, if they stay in hospital, from respiratory support and/or antibiotics). Patients that test negative are treated in line with (4), while patients testing positive are treated in line with (3). Which branches are plausible depends on testing and care practices and resources. COVID, coronavirus disease; MV, mechanical ventilation; O<sub>2</sub>, oxygen; PCR, polymerase chain reaction; RDT, rapid diagnostic test; TCZ, tocilizumab.

called "post-COVID" thereafter. Outcomes around post-COVID remain highly uncertain in LMICs (Section F in S2 Appendix).

## Sensitivity analyses

We used both probabilistic and deterministic sensitivity analysis. In the deterministic sensitivity analysis, we focused on the impact of PCR unavailability (in which case, we compared the use of RDTs alone with no testing), unavailability of MV (assuming conventional oxygen remains available), and different discounting levels, RDT sensitivity, kit and screening costs, linkage between diagnosis and treatment, TCZ costs, side-effects of corticosteroids and treatment impacts on post-COVID (see Section H in S2 Appendix for more details regarding these choices).

We also explored multiple assumptions regarding treatment side effects. Given these are complex, we "bundled" this sensitivity analysis into 3 broad analyses. We considered:

i.  A "naïve" scenario where no side effects are assumed.

ii.  A scenario with only "generic" side effects that do not depend on an individual's disease status. These may be mild/common, e.g., gastro-intestinal effects, or rare/severe: bone fractures, gastro-intestinal bleeding, sepsis, venous thrombosis, and heart failure [13,14] (for corticosteroids) or (for TCZ) tuberculosis (TB) reactivation [15] (S2 Appendix).

iii. A scenario with "disease-specific" effects of COVID treatment. Corticosteroids are suspected to increase mortality in patients with influenza [16] (for full side effects, see Section E in S2 Appendix). Influenza prevalence may be low (but nonzero) outside of an influenza season or outbreak. However, during the influenza season there may be high levels of influenza prevalence among patients with COVID-like illness [17]. We examined scenarios with influenza prevalence among patients with severe COVID-like, non-COVID illness ranging from 0% to 30%.

We finally examined the impact of uncertainty on all model parameters simultaneously in a probabilistic sensitivity analysis (Sections C and D in S3 Appendix).

### Estimates of model parameters and cost-effectiveness

The values of model parameters are listed in Section A in S2 Appendix (including details of parameter distributions and sources) and Section B in S2 Appendix (for country-specific parameters) and available in GitHub (https://github.com/gbn0931/tcov).

### Screening and testing performance

Clinical screening sensitivity and specificity are derived from experts' feedback [8] while RDT and PCR test sensitivity and specificity are based on the literature [18–21]. For RDT sensitivity, the baseline value derives from normative assumptions (80% sensitivity, in line with WHO guidelines [22]). Testing or treatment refusal rates are based on the literature [23–25] and expert consultations [8].

### Health impact of disease and treatment

The impact of COVID treatment was derived from WHO's therapeutic guidelines [9] and a literature review [26]. The probability of severe adverse effects of short-course corticosteroids derives from 2 population-based surveys [13,14]. Disability weights come from Global Burden of Disease (GBD) studies [27]. For COVID years of life lost (YLLs), we used published estimates for 81 countries [28] that we extrapolated to other countries (see methods and results in Section G in S2 appendix). For non-COVID patient deaths, we combined GBD 2019 deaths and YLL data for diseases that may present like COVID and need oxygen supplementation (specifically lower respiratory tract infections (LRTIs), upper respiratory tract infections (URTIs), and TB) and computed an average YLL per non-COVID "COVID-like" death (Section E in S2 Appendix). Health outcomes were not discounted in our primary estimates, while we used a 3% discount rate in sensitivity analysis.

### Costs

Costs are reported in 2021 prices and estimated from a health provider economic perspective using a 3% discounting rate for the costs of treatment side effects lasting more than 1 year. We excluded the economic impact of post-COVID, given the scarcity of evidence from LMICs (Section F in S2 Appendix). We first assumed (Section C in S2 Appendix) that routine screening for respiratory diseases would suffice to identify suspected COVID cases, exploring other assumptions in sensitivity analysis. We extrapolated testing costs across settings from data sourced from the literature as follows: staff costs were assumed to be proportional to overall difference in healthcare personnel salary estimates [29], other non-tradable costs were extrapolated assuming proportionality to GDP per capita (PPP), and US dollar prices for tradeable goods, such as test kits, were considered the same for different countries within a given income group and were inflated using the US GDP deflator when they needed to be adjusted over

time. The costs of COVID hospitalization/days in intensive care unit (ICU)/deaths were extracted from the literature [30]. We used an ingredient-based costing approach, with estimates of the average cost of each input category made from reviewing secondary data sources, to estimate the cost of treatment side effects (Section D in S2 Appendix). Finally, costs in other currencies were translated into dollars using the corresponding year's conversion rate.

## Analysis methods

We calculated net monetary benefits (i.e., incremental benefits times the cost-effectiveness threshold minus incremental costs) using the Ochalek and colleagues [31] country-specific empirical cost-effectiveness thresholds as the willingness to pay for a DALY averted. Assuming the proposed options are affordable, any option with a positive net monetary benefit is more cost-effective than the reference option (neither testing nor treating suspected cases). The most cost-effective option is the one with the highest health benefit for which the incremental cost to incremental health benefit ratio remains below the cost-effectiveness threshold, i.e., the option that maximizes net monetary benefits.

We estimated probabilistic uncertainty by sampling the parameter distributions detailed in S2 Appendix using Latin Hypercube Sampling, simulating 1,000 Monte Carlo samples for each scenario.

Results for different countries were aggregated by income range as parameters are not sufficiently localized to generate accurate individual country results. We present results indicating for which proportion of countries options are cost-effective for a range of SARS-CoV-2 prevalence. We do this as our results do not reflect country context sufficiently to warrant application without consideration of specific contextual factors. Countries wishing to apply these results should combine our main figures with a careful review of our sensitivity analyses that explore the influence of context-specific variables for which we were not able to obtain country-level data.

We explored a range of COVID prevalence between 0% and 30%. S4 Appendix provides individual country graphs without country names to illustrate the variability in country situations within the same income level. The model was coded in R version 4.2.2 and is available (alongside associated datasets) at https://github.com/gbn0931/tcov.

## Results

### Treatment scenario 1: Only corticosteroids are available

**Baseline scenario.** The most common disease that may present with COVID-like symptoms and on which corticosteroids may have substantial disease-specific effects, notably a large increase in mortality [16], is influenza. We therefore present results for low and high influenza prevalence. Fig 2 presents the share of countries at each income level for which different testing options are likely to be the most cost-effective when influenza prevalence among patients with severe, COVID-like, non-COVID disease is 1%. At low COVID prevalence, treating presenting patients as if they did not have COVID is the most cost-effective option in low-income countries. However, as SARS-CoV-2 prevalence increases, testing with RDTs becomes the most cost-effective option in the majority of (though not all) low-income countries. Meanwhile, at very high prevalence, presumptive treatment is the most cost-effective option. In upper middle-income countries, however, testing is always cost-effective, even at very low COVID prevalence. Furthermore, at high SARS-CoV-2 prevalence, negative RDT test results should be confirmed with PCR.

In Fig 3, we present the same results assuming a higher prevalence of 10% influenza among patients with severe COVID-like, non-COVID disease. In low-income countries, at low

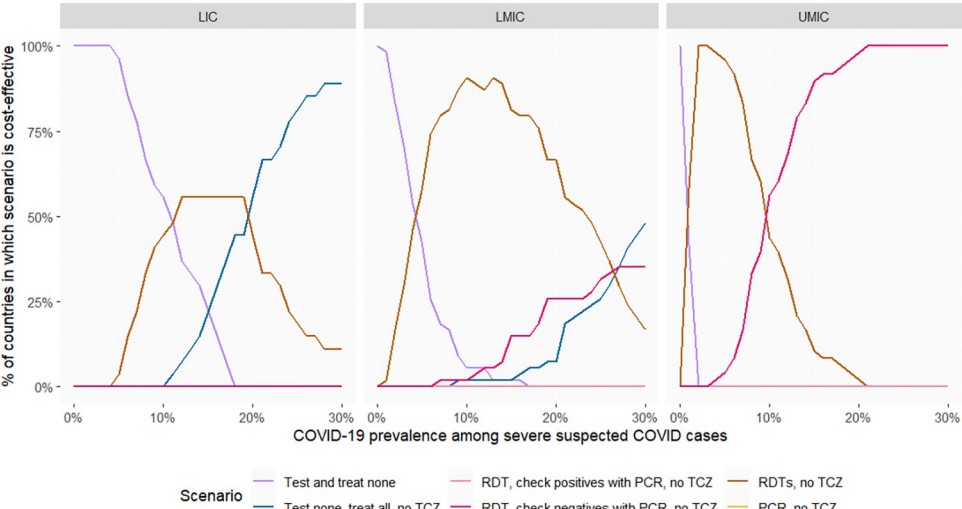

**Fig 2. Proportion of countries in which a given testing option is the most cost-effective (with 1% influenza prevalence among COVID-like, non-COVID patients).** The "most cost-effective option" is the option most likely to be cost-effective based on simulation of 1,000 parameter sets. PCR, polymerase chain reaction; RDT, rapid diagnostic tests; TCZ, tocilizumab; LICs, low-income countries (27 countries); LMICs, lower-middle-income countries (54 countries); UMICs, upper-middle-income countries (48 countries).

prevalence, the most cost-effective option remains not to test, and to treat patients with COVID-like illnesses as if they did not have COVID, while at high prevalence, RDT testing is the most cost-effective option. The most cost-effective options are similar in lower-middle-income countries, except that the COVID prevalence threshold beyond which RDT testing is the most cost-effective option is lower. Meanwhile, in most upper-middle-income countries, testing algorithms with high specificity (confirming RDT-positives with PCR or, in some cases, PCR alone without RDT) become the most cost-effective options at low SARS-CoV-2

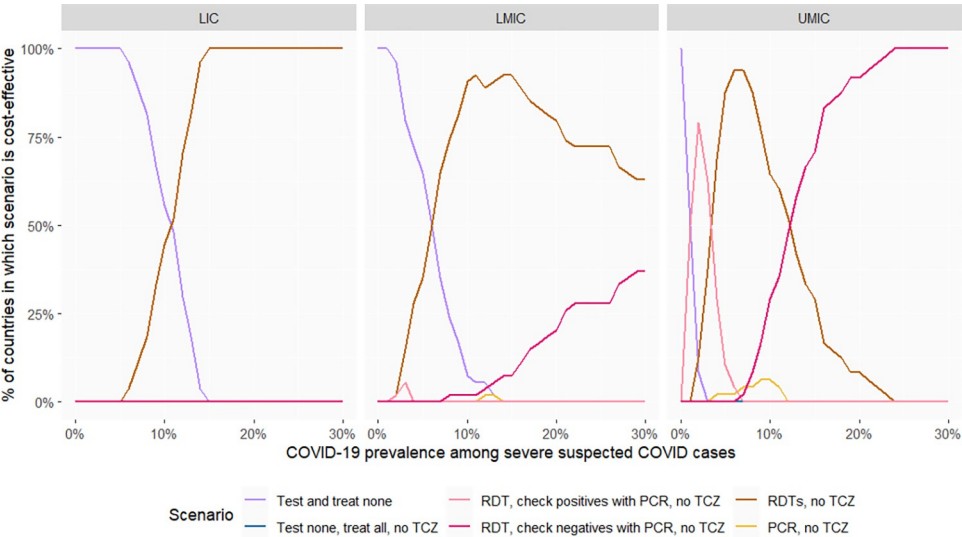

**Fig 3. Proportion of countries in which a given testing option is the most cost-effective (with 10% influenza prevalence among COVID-like, non-COVID patients).** The "most cost-effective option" is the option most likely to be cost-effective based on simulation of 1,000 parameter sets. PCR, polymerase chain reaction; RDT, rapid diagnostic tests; TCZ, tocilizumab; LICs, low-income countries (27 countries); LMICs, lower-middle-income countries (54 countries); UMICs, upper-middle-income countries (48 countries).

prevalence as the relative costs of treating false positives increase, while confirming test-negatives with PCR is most cost-effective at high SARS-CoV-2 prevalence.

We explore the impact of varying influenza prevalence among patients with severe COVID-like, non-COVID disease from near-zero to 30% in Section A in S3 Appendix. Broadly, above 2.5% influenza prevalence, RDT testing starts to become more cost-effective than presumptive treatment (for some SARS-CoV-2 prevalence levels at least) in all LMICs, and at 5% influenza prevalence, presumptive treatment is never cost-effective. Meanwhile, confirming positive test results at low SARS-CoV-2 prevalence becomes cost-effective in upper-middle-income countries and some lower-middle-income countries (those with the highest cost-effectiveness threshold) when influenza prevalence in patients with severe COVID-like, non-COVID illness crosses a threshold inversely proportional to a country's willingness to pay, the median threshold being around 10%.

## Sensitivity analyses

**Impact of different approaches to corticosteroid side effects.** If any consideration of corticosteroid side effects is removed, testing (with any test algorithm) is never the most cost-effective option (Section C in S3 Appendix), as presumptive corticosteroid treatment costs less than testing while producing equal or greater health benefits (no false negatives missed, no side effects in false positives). At low SARS-CoV-2 prevalence, it is most cost-effective to treat all suspected cases as not having COVID, while at higher prevalence levels, presumptively treating all suspected cases with corticosteroids is most cost-effective.

If only the generic side effects of corticosteroids (Section D in S2 Appendix) are considered, RDT testing is still not the most cost-effective option in most low-income countries and contexts. However, testing has a positive net monetary benefit (see Section A in S3 Appendix) when SARS-CoV-2 prevalence is high. RDT testing alone becomes the most cost-effective option in upper-middle- and most lower-middle-income countries. At high prevalence levels, confirming negative test results with PCR is the most cost-effective option in upper-middle-income countries.

**Other sensitivity analyses.** We summarize these results in Fig 4, which provides the share of countries in which testing is cost-effective for at least 1 value of SARS-CoV-2 prevalence. For example, if RDT test kits prices were at $6.20, the share of countries in which testing would be cost-effective would decrease by 12 percentage points. On the other hand, if RDT test kit prices were at $0.60, the share of countries in which testing would be cost-effective would increase by 8 percentage points. For these analyses, we assume that influenza-related side effects are present, with an influenza prevalence of 1% at baseline. The most influential parameter is influenza prevalence, followed by RDT sensitivity and test kit costs. PCR availability does not have any impact on the number of countries in which RDT is cost-effective, although it affects which testing scenario is preferable (see Section C in S3 Appendix for detailed figures associated with each of the scenarios presented in Fig 4).

## Treatment scenario 2: Corticosteroids and IL-6 receptor blockers (TCZ) available

**Baseline scenario.** For simplicity, the results presented for this section assume 1% influenza prevalence. Providing TCZ presumptively without testing was not cost-effective using a medium price of $861·5 per treatment course in any setting. In low-income countries, testing and providing TCZ was also never the most cost-effective option. Testing and treatment with TCZ was the most cost-effective option in all upper-middle-income countries. Testing before using TCZ on positive patients was cost-effective in a little under half of lower-middle-income countries at high SARS-CoV-2 prevalence (Fig 5).

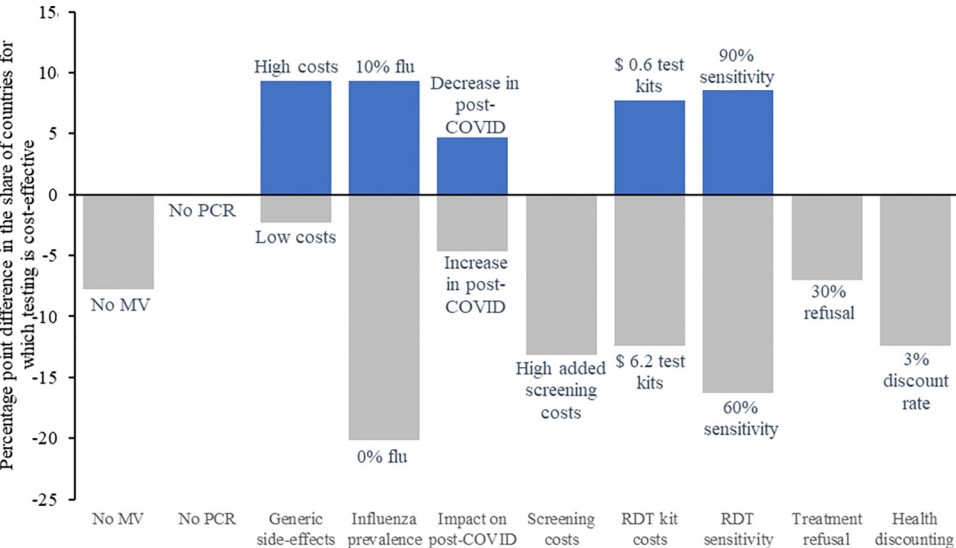

**Fig 4. Change in the proportion of countries for which testing is the most cost-effective option\*.** \* For at least 1 SARS-CoV-2 prevalence value. The size of each bar represents the difference in percentage points with the baseline scenario (baseline value for all parameters and 1% influenza among severe COVID-like patients) using primary estimates for all parameters. COVID, coronavirus disease; MV, mechanical ventilation; PCR, polymerase chain reaction; RDT, rapid diagnostic tests.

## Sensitivity analyses

**Impact of TCZ price.** Section B in S3 Appendix shows the change in results when TCZ price is varied between $411 and $1,207 per treatment course. Even at low ($411 per treatment course) prices, testing to inform TCZ use is never cost-effective in any low-income country. As

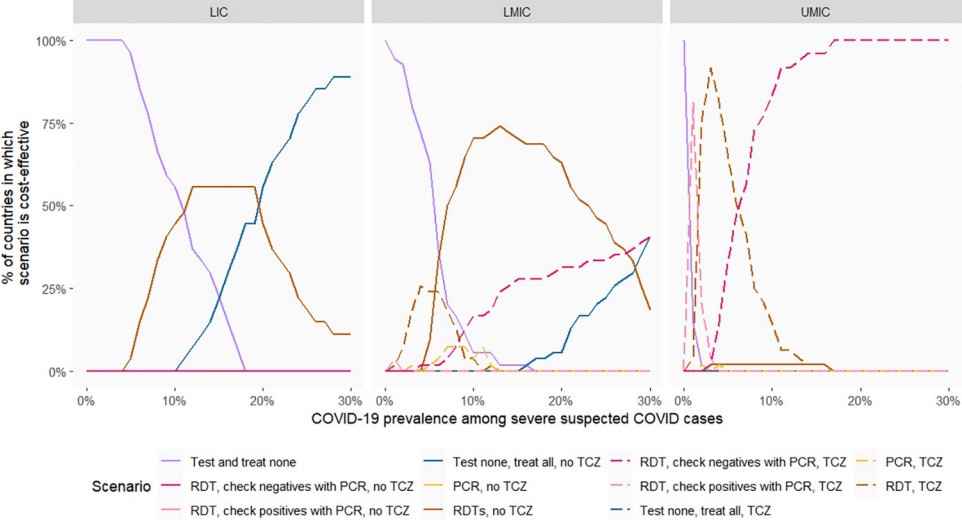

**Fig 5. Most cost-effective option\* when TCZ is available.** \* The "most cost-effective option" is the option most likely to be cost-effective based on simulation of 1,000 parameter sets. PCR, polymerase chain reaction; RDT, rapid diagnostic tests; TCZ, tocilizumab. Dotted lines are used to represent options in which TCZ is used, whereas full lines represent options in which it is not. LICs, low-income countries (27 countries); LMICs, lower-middle-income countries (54 countries); UMICs, upper-middle-income countries (48 countries).

the price of therapeutics may evolve over time, however, we also estimated the maximum cost of a TCZ treatment course for it to remain cost-effective at different SARS-CoV-2 prevalence values. Section B in S3 Appendix shows the median of these values across low-, lower-middle-, and upper-middle-income countries. The curves increase rapidly with SARS-CoV-2 prevalence then reach a plateau. In low-income countries, the cost of TCZ treatment should be at most between $162 and 406 (median: $278) for the option of RDT followed by TCZ treatment of positives to be cost-effective at least at some SARS-CoV-2 prevalence levels. Those figures are $625, range [$287, $5,286] and $2,618, range [$1,108, $8,344] in lower-middle- and upper-middle-income countries, respectively.

**Other sensitivity analyses.** We also explored our results for the different alternative scenarios described in Methods. Section C in S3 Appendix presents the most cost-effective option for each of these scenarios, by country income level and COVID prevalence. The impacts of the different scenarios on the cost-effectiveness of testing when TCZ is available and when it is not are similar.

Finally, Section D in S3 Appendix presents a tornado graph showing the 25 model parameters with the largest influence on the net health benefit of testing. Our analysis confirms that the main drivers of variability in net health benefit are a country's income level (reflected in differences in hospitalization costs) and SARS-CoV-2 prevalence. The net health benefit of RDTs is sensitive to COVID case-fatality risks, suggesting that variations in the viral variants and/or population immunity could be important drivers of changes in model results.

## Discussion

Our analysis shows that COVID testing among patients with severe, COVID-like illness can be cost-effective in all LMICs depending on SARS-CoV-2 and influenza prevalence among COVID-like patients.

SARS-CoV-2 prevalence varies over time, so developing universal testing policies based on prevalence may be difficult. However, our expert consultation [8] suggests that clinicians are likely to consider local SARS-CoV-2 prevalence at a given time, and combine this with clinical symptoms to determine the "index of suspicion" for a patient. Given this, one way to consider prevalence is for countries to define clinical screening algorithms that are more specific than commonly used algorithms hence lead to the identification of patients with a higher index of suspicion, particularly in low-income settings where testing is only cost-effective for such patients.

With this in mind, our findings imply that in low-income countries, suspected cases with a low index of suspicion may be treated as non-COVID, whereas those with a high index of suspicion should be tested with RDTs to inform treatment decision. In lower-middle-income countries, only patients with a much lower index of suspicion should be treated as non-COVID, and all other suspected cases should be tested. In the better resourced of those countries, when the index of suspicion (pre-test) is high, patients' negative RDT results may be confirmed with PCR, whereas when the index of suspicion is low and influenza prevalence very high, positive RDT results may be confirmed with PCR. Finally, in upper-middle-income countries, all suspected cases should be tested and those with a high index of suspicion (pre-test) and a negative RDT result confirmed with PCR. Furthermore, during influenza outbreaks, positive test results in a patient with a low index of suspicion for COVID (pre-test) should be confirmed with PCR.

Testing may serve to guide TCZ use in upper-middle- and some lower-middle-income countries, but TCZ is too costly, at current prices, in low-income countries. However, it is important to note that the most cost-effective option is sensitive to influenza prevalence,

**Table 1. Most cost-effective option depending on country income level, influenza, and COVID prevalence among patients with severe COVID-like illness.**

| Country type | Influenza prevalence (among COVID-like non-COVID patients) | Maximum TCZ cost for its use to be cost-effective | Most cost-effective testing option with low SARS-CoV-2 prevalence | Most cost-effective testing option with high SARS-CoV-2 prevalence |
|---|---|---|---|---|
| Low-income | Low (around 1%) | $162 to $406 (median $278) depending on the country | Do not test (treat as non-COVID) | Test (RDT)* |
| Low-income | High (around 10%) | | Do not test (treat as non-COVID) | Test (RDT) |
| Lower-middle-income | Low (around 1%) | $287 to $5,286 (median $625) depending on the country | Test (RDT) except at very low COVID prevalence | Test (RDT)** |
| Lower-middle-income | High (around 10%) | | Test (RDT) except at very low COVID prevalence*** | Test (RDT)** |
| Upper-middle-income | Low (around 1%) | $1,108 to $8,344 (median $2,618) depending on the country | Test (RDT) | Test (RDT), confirm negatives with a PCR test |
| Upper-middle-income | High (around 10%) | | Test (RDT), confirm positives with a PCR test | Test (RDT), confirm negatives with a PCR test |

* Treating presumptively may be more cost-effective than testing when there is no or virtually no influenza, but if influenza prevalence or country-specific thresholds are uncertain, testing is safer.

** If the cost-effectiveness threshold is above $950, it may be more cost-effective to confirm negatives with PCR than just test with RDTs.

*** If the cost-effectiveness threshold is above $800 and influenza prevalence very high, confirming RDT-positives with PCR may be the most cost-effective option. For more discussion of influenza and SARS-CoV-2 prevalence thresholds, see S3.1.3 and S3.1.4.

COVID, coronavirus disease; PCR, polymerase chain reaction; RDT, rapid diagnostic test; SARS-CoV-2, Severe Acute Respiratory Syndrome Coronavirus 2; TCZ, tocilizumab.

which may not be well known in countries without laboratory-based acute respiratory infection surveillance (Table 1).

Our results align with current practice [8] in many contexts, such as regarding PCR confirmation of negative RDTs in patients with a high index of suspicion. However, our analysis differs from prior LMIC models [10,11] in several respects because of the breadth of testing/treatment options and country contexts considered and the inclusion of unintended treatment effects. Notably, we were able to demonstrate that, when corticosteroids are the only available therapeutics, the cost-effectiveness of testing is driven by the need to avoid their side effects. We show that, in certain epidemiological contexts (influenza epidemics), the side effects of corticosteroids may motivate the choice of testing algorithms (e.g., confirming RDT-positives with PCR) which may suggest that the common practice [8] of taking test-positives at face value needs modification.

The main strength of this study is its comprehensiveness, particularly with regard to the assessment of the unintended effects of COVID treatment. However, we have had to make some key assumptions and simplifications. We have only assessed the health benefit of testing for the individual being tested, assuming no indirect impact due to reduced transmission. However, where isolation of confirmed COVID cases is feasible, testing may support a reduction in nosocomial COVID prevalence. Testing may inform the behavior of cases' contacts, reducing their onward transmission into the community. Testing may also help reduce antibiotic resistance or contribute to surveillance, helping, for example, identify new variants.

Another key simplification is that we have only quantified the impact of treatment for COVID, or its absence, on COVID and non-COVID patients. In doing so, we have ignored the impact of treating a COVID patient erroneously with therapeutics specific to other diseases or of failing to treat a non-COVID patient who tests positive for COVID (false positive) with the appropriate therapeutics. The inclusion of these considerations would likely have increased the impact of testing algorithms that provide more accurate test results. Another underlying

assumption is that, among patients with severe or critical illness, care-seeking behaviors are driven primarily by symptom severity and not substantially altered by the presence/absence of SARS-COV-2 testing. This may not be accurate. However, these care-seeking behaviors are complex and poorly understood [8] and a revised model accounting for change in such behaviors depending on the testing scenario considered would require prior country-specific data collection.

In addition, when calculating the impact of testing, we have assumed instantaneous availability of test results. This is a good approximation with RDTs but less accurate with PCR. At the beginning of the pandemic, PCR test results in some contexts were so delayed they could not be used to inform clinical decisions. However, test results are now generally obtained within 24 h [8], so our approximation is likely more accurate. Some clinicians may also test but ignore the test results: we have ignored this phenomenon because of difficulties in quantifying the factors that enter into such decisions. Finally, in discussing the different scenarios, we have implicitly assumed some knowledge of the general level of COVID and influenza prevalence in the target populations. Some level of testing for surveillance may be useful to inform this, but it is beyond the scope of our study to make specific recommendations about surveillance.

Our analysis suggests that SARS-CoV-2 testing can still be cost-effective to support the treatment of patients in LMICs with severe COVID-like illness, provided testing can be targeted at patients with a sufficiently high index of suspicion, particularly in lower-income settings. Country decision-makers may consider results for the country income group relevant to them, variability within this group as well as the results of the sensitivity analyses, alongside other factors such as any locally relevant research, their own judgment, feasibility, or political factors to inform local decision-making.

Our analysis also highlights the importance of considering the negative impact of corticosteroids on severe influenza outcomes. Further research on understanding the likelihood and magnitude of such impacts is needed to further improve testing and treatment algorithms for acute respiratory illness.

In conclusion, we presented a detailed analysis of the value of testing to inform treatment in health facilities in LMICs. We illustrated a complex picture of real-world practice and relationships with treatment pathways, SARS-CoV-2 prevalence, and treatment cost. Our results show that different policies may be cost-effective depending on the income level of countries. They also highlight the importance of further research into some of the largest driver of uncertainty in the model: the side effects of corticosteroids and particularly their impacts on non-COVID cases such as influenza.

## Supporting information

**S1 Appendix. Context and checklists.**
(DOCX)

**S2 Appendix. Parameters.**
(DOCX)

**S3 Appendix. Additional figures and tables—most cost-effective options in different scenarios.**
(DOCX)

**S4 Appendix. Country-specific graphs.**
(DOCX)

## Author Contributions

**Conceptualization:** Gabrielle Bonnet, Mark Jit, Anna Vassall.

**Formal analysis:** Gabrielle Bonnet.

**Funding acquisition:** Elizabeth L. Corbett.

**Investigation:** Gabrielle Bonnet, John Bimba, Chancy Chavula, Harunavamwe N. Chifamba, Titus H. Divala, Andres G. Lescano, Mohammed Majam, Danjuma Mbo, Auliya A. Suwantika, Marco A. Tovar, Pragya Yadav, Obinna Ekwunife, Collin Mangenah, Lucky G. Ngwira, Mark Jit, Anna Vassall.

**Methodology:** Gabrielle Bonnet, Elizabeth L. Corbett, Mark Jit, Anna Vassall.

**Software:** Gabrielle Bonnet.

**Supervision:** Elizabeth L. Corbett, Mark Jit, Anna Vassall.

**Validation:** Mark Jit, Anna Vassall.

**Visualization:** Gabrielle Bonnet, Elizabeth L. Corbett, Mark Jit, Anna Vassall.

**Writing – original draft:** Gabrielle Bonnet.

**Writing – review & editing:** Gabrielle Bonnet, John Bimba, Chancy Chavula, Harunavamwe N. Chifamba, Titus H. Divala, Andres G. Lescano, Mohammed Majam, Danjuma Mbo, Auliya A. Suwantika, Marco A. Tovar, Pragya Yadav, Obinna Ekwunife, Collin Mangenah, Lucky G. Ngwira, Elizabeth L. Corbett, Mark Jit, Anna Vassall.

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
