## [Editor Report · Decision Letter 0]

22 Jan 2024

Dear Dr Bonnet, 

Thank you for submitting your manuscript entitled "To test or not to test? A cost-effectiveness analysis of COVID rapid diagnostic tests for severe/critical patients in Low-and-Middle Income Countries" for consideration by PLOS Medicine.

Your manuscript has now been evaluated by the PLOS Medicine editorial staff and I am writing to let you know that we would like to send your submission out for external peer review.

Please re-submit your manuscript within two working days, i.e. by Jan 24 2024.

Feel free to email me at aschaefer@plos.org if you have any queries relating to your submission.

Kind regards,

Alexandra Schaefer, PhD

Associate Editor

PLOS Medicine

---

## [Decision Letter · Decision Letter 1]

28 Feb 2024

Dear Dr. Bonnet,

Thank you very much for submitting your manuscript "To test or not to test? A cost-effectiveness analysis of COVID rapid diagnostic tests for severe/critical patients in Low-and-Middle Income Countries" (PMEDICINE-D-24-00195R1) for consideration at PLOS Medicine. 

Your paper was evaluated by an associate editor and discussed among all the editors here. It was also discussed with an academic editor with relevant expertise, and sent to independent reviewers, including a statistical reviewer. The reviews are appended at the bottom of this email and any accompanying reviewer attachments can be seen via the link below:

[LINK]

In light of these reviews, I am afraid that we will not be able to accept the manuscript for publication in the journal in its current form, but we would like to consider a revised version that addresses the reviewers' and editors' comments. Obviously we cannot make any decision about publication until we have seen the revised manuscript and your response, and we plan to seek re-review by one or more of the reviewers. 

Please use the following link to submit the revised manuscript: https://www.editorialmanager.com/pmedicine/

We expect to receive your revised manuscript by Mar 20 2024. However, if this deadline is not feasible, please contact me by email, and we can discuss a suitable alternative.

Don't hesitate to contact me directly with any questions (aschaefer@plos.org). If you reply directly to this message, please be sure to 'Reply All' so your message comes directly to my inbox.

We look forward to receiving your revised manuscript.

Sincerely,

Alexandra Schaefer, PhD

PLOS Medicine

plosmedicine.org

Requests from the editors (Please note: not all will apply to your paper, but please check each item carefully):

ACADEMIC EDITOR COMMENTS

The focus should be on making the manuscript comprehensible to the majority of readers. The reviewers all strongly recommended this. As I read through the results, I found myself looking for the summary table that would tell me the outcomes of all the scenarios in simple terms. I think the reliance on Figures 2 and 3 to convey the most important results might be challenging, as most readers will not understand these figures, or even be inclined to try to decipher them. I also found that I still don't know what the results mean for any actual, individual country or implementer. Specific case study examples would help a lot.

The comments that struck me as most important to respond to are:

Reviewer 1, second paragraph under "General" heading ("My main concern..."). This is well stated, and I agree entirely.

Reviewer 2, second paragraph ("My first issue...), which reiterates the concerns of Reviewer 1, and the comment: ""Our analysis shows that the most cost-effective way to use RDTs to inform COVID treatment depends on country income, SARS-COV-2 prevalence, and flue prevalence". These are three factors that you picked out to analyse ex ante in this study - are there any other important factors that you weren't able to capture in your analysis that are noteworthy?"

Reviewer 3, "The cost of a test kit is now around ≤$2 (landed price)- how does this alter or streamline the recommendations from this paper?" (I am also interested in the implications for antimicrobial stewardship, but that might be beyond the scope of this manuscript.)

EDITORIAL COMMENTS

The Editors concur with the Academic Editor that the study needs to be streamlined and the message simplified to be accessible to a broad audience. Regarding the specific case study examples, we are unsure whether this might create too much of a focus on specific countries and detract from the overall message. However, if you feel this would strengthen your manuscript while making it more accessible, we leave it up to you to decide whether to include these examples.

The Editors also discussed whether your analysis should be expanded to include broader societal costs/wider economic considerations, including socioeconomic impacts, e.g. through an extended cost-effectiveness analysis (ECEA). We would be interested to hear your thoughts on this and whether this would be a feasible approach that would be in keeping with the purpose of your study, taking into account the need to make the study more accessible to the average reader.

GENERAL COMMENTS

1) Please cite the reference numbers in square brackets. Citations should be preceding punctuation.

2) Please temper claims of primacy of results by stating, "to our knowledge" or something similar.

FINCANCIAL DISCLOSURE

The funding statement should include: specific grant numbers, initials of authors who received each award, URLs to sponsors’ websites. Also, please state whether any sponsors or funders (other than the named authors) played any role in study design, data collection and analysis, the decision to publish, or preparation of the manuscript. If they had no role in the research, include this sentence: “The funders had no role in study design, data collection and analysis, decision to publish, or preparation of the manuscript.”

COMPETING INTEREST

All authors must declare their relevant competing interests per the PLOS policy, which can be seen here:

https://journals.plos.org/plosmedicine/s/competing-interests

For authors with ties to industry, please indicate whether any of the interests has a financial stake in the results of the current study.

TITLE

Please revise your title according to PLOS Medicine's style. Your title must be nondeclarative and not a question. It should begin with main concept if possible. "Effect of" should be used only if causality can be inferred, i.e., for an RCT. Please place the study design ("A randomized controlled trial," "A retrospective study," "A modelling study," etc.) in the subtitle (ie, after a colon). Please remove “To test or not to test?” from the study title.

ABSTRACT

1) Please structure your abstract using the PLOS Medicine headings (Background, Methods and Findings, Conclusions). Please combine the Methods and Findings sections into one section, “Methods and findings”.

2) PLOS Medicine requests that main results are quantified with 95% CIs as well as p values. When reporting p values please report as p<0.001 and where higher as the exact p value p=0.002, for example. For the purposes of transparent data reporting, if not including the aforementioned please clearly state the reasons why not. When a p value is given, please specify the statistical test used to determine it. 

3) Throughout, suggest reporting statistical information as follows to improve clarity for the reader “22% (95% CI [13%,28%]; p</=)”. Please be sure to define all numerical values at first use. Please amend throughout the abstract and main manuscript. Please note the use of commas to separate upper and lower bounds, as opposed to hyphens as these can be confused with reporting of negative values.

4) Please ensure that all numbers presented in the abstract are present and identical to numbers presented in the main manuscript text.

5) Please include the study design, population and setting, number of participants, years during which the study took place, length of follow up, main outcome measures.

6) Please include the important dependent variables that are adjusted for in the analyses.

7) Please define all abbreviations including those for statistical reporting at first use.

8) Abstract Background: The final sentence should clearly state the study question.

9) In the last sentence of the Abstract Methods and Findings section, please describe the main limitation(s) of the study's methodology.

AUTHOR SUMMARY

At this stage, we ask that you include a short, non-technical Author Summary of your research to make findings accessible to a wide audience that includes both scientists and non-scientists. The authors summary should consist of 2-3 succinct bullet points under each of the following headings:

• Why Was This Study Done? Authors should reflect on what was known about the topic before the research was published and why the research was needed.

• What Did the Researchers Do and Find? Authors should briefly describe the study design that was used and the study’s major findings. Do include the headline numbers from the study, such as the sample size and key findings.

• What Do These Findings Mean? Authors should reflect on the new knowledge generated by the research and the implications for practice, research, policy, or public health. Authors should also consider how the interpretation of the study’s findings may be affected by the study limitations. In the final bullet point of ‘What Do These Findings Mean?’, please describe the main limitations of the study in non-technical language.

Author Summary should immediately follow the Abstract in your revised manuscript. This text is subject to editorial change and should be distinct from the scientific abstract. Please see our author guidelines for more information: https://journals.plos.org/plosmedicine/s/revising-your-manuscript#loc-author-summary

METHODS AND RESULTS

1) Please ensure that the study is reported according to the CHEERS guideline, and include the completed CHEERS checklist as Supporting Information. Please add the following statement, or similar, to the Methods: "This study is reported as per the Consolidated Health Economic Evaluation Reporting Standards 2022 (CHEERS2022) Statement (S1 Checklist)."

2) PLOS Medicine requests that main results are quantified with 95% CIs as well as p values. We suggest reporting statistical information as detailed above – see under ABSTRACT

3) Please present numerators and denominators for percentages (at least in the Tables [not necessarily each time they're mentioned]).

DISCUSSION

Please present and organize the Discussion as follows: a short, clear summary of the article's findings; what the study adds to existing research and where and why the results may differ from previous research; strengths and limitations of the study; implications and next steps for research, clinical practice, and/or public policy; one-paragraph conclusion.

FIGURES

For all Figures, please ensure that you have complied with our figures requirements http://journals.plos.org/plosmedicine/s/figures.

1) Please provide titles and legends for all figures (including those in Supporting Information files).

2) Please consider avoiding the use of red and green in order to make your figure more accessible to those with color blindness.

3) Please in the figure legend/description, define abbreviations used in each figure (including those in Supporting Information files).

4) Please define the meaning of all dots, lines and bars in the captions/footnotes.

TABLES

1) Please provide titles and legends for all tables (including those in Supporting Information files).

2) Please define all abbreviations used in the table below each table (including those in Supporting Information files). 

SUPPLEMENTARY MATERIAL

1) For supplementary figures and tables, please see the general comments under TABLES and FIGURES (color, abbreviations, titles, descriptions, etc.) and amend accordingly.

2) We suggest reporting statistical information as detailed above – see under ABSTRACT. Please be sure to define all numerical values.

3) As for the main manuscript, please indicate whether analyses are adjusted to help facilitate transparent data reporting please also detail the factors adjusted for and present the unadjusted analyses for comparison. If not, please clearly state the reasons why not.

4) Please revise the references in the supplementary material according to comments detailed below – see under REFERENCES.

5) Please cite your Supporting Information as outlined here: https://journals.plos.org/plosmedicine/s/supporting-information

REFERENCES

1) PLOS uses the numbered citation (citation-sequence) method and first six authors, et al.

2) Please ensure that journal name abbreviations match those found in the National Center for Biotechnology Information (NCBI) databases (http://www.ncbi.nlm.nih.gov/nlmcatalog/journals), and are appropriately formatted and capitalised.

3) Where website addresses are cited, please specify the date of access (e.g. [accessed: 12/06/2023]).

4) Please also see https://journals.plos.org/plosmedicine/s/submission-guidelines#loc-references for further details on reference formatting. 

Comments from the reviewers:

Reviewer #1: Review: "To test or not to test? A cost-effectiveness analysis of COVID rapid diagnostic tests for severe/critical patients in Low-and-Middle Income Countries"

PLOS Medicine

2/6/24

General

The authors present a novel study assessing the cost-effectiveness of COVID RDTs for severe/critical patients in LMIC settings. Using a dizzying array of statistical modeling based on published and/or publicly-available data, the authors provide a tremendous number of "real-world" scenarios involving infection prevalence rates, treatment side effect rates, and medication availability rates (amongst other factors) to inform their analyses. Strong points include how widely representative the author list is, both by geography and by country income level, as well as the intensely comprehensive nature of the analyses. Overall, I believe that this manuscript has intrinsic value and is worthy of publication. I applaud the authors for their clearly hard work. 

My main concern is that the level of comprehensiveness may detract from the potential impact of the paper. In other words, are the authors trying to accomplish too much all at once and, if so, could they make their same points using fewer subgroup analyses? Is every possible scenario included in this paper necessary? Trying to accomplish everything in one fell swoop creates very dense text and innumerable, highly complex figures, each with multiple variables that require intense concentration to process, that limit the reader's ability to digest the complexity of the analysis in any easy way. The authors would do well by simplifying the analysis (if possible) and limiting the figures—or, perhaps—adding a simple table—that more clearly and easily highlights the most salient findings of their study. 

Abstract

-The Findings and Interpretation sections, at first read, are confusing as the authors bounce back and forth between mentioning COVID, influenza and steroids. A better organized abstract that explains earlier on how steroids and influenza are relevant to this paper would make it easier to understand. 

Introduction

-No specific concerns. 

Method

-Line 97: "We applied a health provider perspective." This line appears to contradict line 190 and the abstract, which indicates that the authors used "a healthcare payer perspective and net monetary benefit approach." Please clarify. 

-Line 105: Please help me to understand how estimates for testing and other factors can be made

---

## [Decision Letter · Decision Letter 2]

31 May 2024

Dear Dr. Bonnet,

Thank you very much for re-submitting your manuscript "Cost-effectiveness of COVID rapid diagnostic tests for severe/critical patients in low-and-middle income countries, a modelling study" (PMEDICINE-D-24-00195R2) for review by PLOS Medicine.

Thank you for your detailed response to the editors' and reviewers' comments. I have discussed the paper with my colleagues and the academic editor, and it has also been seen again by two of the original reviewers. The changes made to the paper were satisfactory to the reviewer. As such, we intend to accept the paper for publication, pending your attention to the editorial comments below in a further revision. When submitting your revised paper, please once again include a detailed point-by-point response to the editorial comments.

[LINK]

In revising the manuscript for further consideration here, please ensure you address the specific points made by each reviewer and the editors. In your rebuttal letter you should indicate your response to the reviewers' and editors' comments and the changes you have made in the manuscript. Please submit a clean version of the paper as the main article file. A version with changes marked must also be uploaded as a marked up manuscript file. Please also check the guidelines for revised papers at http://journals.plos.org/plosmedicine/s/revising-your-manuscript for any that apply to your paper. 

We ask that you submit your revision within 1 week (Jun 07 2024). However, if this deadline is not feasible, please contact me by email, and we can discuss a suitable alternative.

Please do not hesitate to contact me directly with any questions (atosun@plos.org). If you reply directly to this message, please be sure to 'Reply All' so your message comes directly to my inbox.

We look forward to receiving the revised manuscript.

Sincerely,

Alexandra Tosun, PhD

Associate Editor

PLOS Medicine

plosmedicine.org

Requests from Editors:

ABSTRACT

1) l.54: Please define “PCR” at first use.

2) l.69: Please write “IL-6” in full.

AUTHOR SUMMARY

1) l.85: Please define ‘LMIC’ at first use.

2) l.96: Please write ‘RDT’ in full.

3) l.98: Please write ‘PCR’ in full.

INTRODUCTION

l.69: Please define “IL-6” at first use.

METHODS AND RESULTS

1) l.139: We suggest that you include a reference here, or where you feel it is appropriate, to the list of countries included. 

2) Figure 1: What do the boxes with a dashed outline signify?

3) l.228: Please check carefully whether S3.4 Appendix should be referenced here and not S3.3 Appendix.

4) l.257: Please define “ICU” at first use or write in full. 

5) Figure 2/3/5: In the figure description, please define “LIC”, “LMIC”, and “UMIC”. Also, we suggest including the numbers (of 129) that were included in the three different country groups (LIC, LMIC, UMIC). 

6) Figure 4: Since you are displaying percentage points and not percentages, please remove the unit from the y-axis.

7) ll.345-351: For better comprehension of the figure, we suggest detailing one example, e.g. test kit costs, with the relevant numbers. For example (assuming we have interpreted the figure and data correctly): If RDT test kits prices were at $6.20, the share of countries in which testing would be cost-effective would decrease by 15 percentage points. On the other hand, if RDT test kit prices were at $0.60, the share of countries in which testing would be cost-effective would increase by 7.5 percentage points. 

8) ll.350-351: “(see S3.2 Appendix for detailed figures associated with each of the scenarios presented in Fig 4).” – should this be S3.3 Appendix? Please carefully revise the manuscript for referencing the relevant supplementary materials and make sure that at minimum the main appendices are all referenced throughout the main manuscript (i.e. second level: S1.1, S1.2, etc.).

DISCUSSION

1) Table 1: In the table, you have marked the footnotes as *, **, *** (1,2, 3), whereas in the description you have listed the asterisks as **, ***, **** (2,3,4) - please revise. Please define “TCZ”.

2) Please remove the “Conclusions” subheading.

REFERENCES

Where website addresses are cited, when specifying the date of access, please use the word “accessed” instead of “cited” (e.g. [accessed: 10/04/2024]).

SOCIAL MEDIA

To help us extend the reach of your research, please provide any X (formerly known as Twitter) handle(s) that would be appropriate to tag, including your own, your co-authors’, your institution, funder, or lab. Please enter in the submission form any handles you wish to be included when we post about this paper.

Comments from Reviewers:

Reviewer #2: Thank you to the authors for resubmitting this paper.

The authors have sufficiently responded to each of my individual concerns (and look to addressed the concerns of the other reviewers too). The addition of the table with an overarching summary of the primary results is much appreciated, as is the reduction in the CEAC type figure to three panels - hopefully both of these additions will make it easier for the general reader to interpret the results. 

Reviewer #3: Fantastic update to this paper and response to our comments and suggestions- well done :).

[LINK]

General Editorial Requests

---

## [Editor Report · Decision Letter 3]

19 Jun 2024

Dear Dr Bonnet, 

On behalf of my colleagues and the Academic Editor, Sydney Rosen, I am pleased to inform you that we have agreed to publish your manuscript "Cost-effectiveness of COVID rapid diagnostic tests for severe/critical patients in low-and-middle income countries, a modelling study" (PMEDICINE-D-24-00195R3) in PLOS Medicine.

I appreciate your thorough responses to the reviewers' and editors' comments throughout the editorial process. We look forward to publishing your manuscript, and editorially there are only a few remaining minor stylistic/presentation points that should be addressed prior to publication. We will carefully check whether the changes have been made. If you have any questions or concerns regarding these final requests, please feel free to contact me at atosun@plos.org.

Please see below the minor points that we request you respond to:

1) l.50: As we prefer the use of patient-centered language, please change "severe patients" to "patients with severe symptoms" (or "patients with severe illness") or similar. Please revise the title and main manuscript accordingly.

2) ll.52-60: The Abstract does not currently mention that your analysis is based on a decision tree model. Please include this detail in the Methods and Findings section of the Abstract (e.g. as in the first bullet under "What did the researchers do and find?" in the Author Summary).

3) l.69: Since there is no use of the abbreviation 'IL-6' in the Abstract after its introduction on line 69, please remove it.

4) Title: Please place the study design after a colon (i.e., “Cost-effectiveness of COVID rapid diagnostic tests for patients with severe symptoms in low- and-middle income countries: A modelling study”).

PRESS

Sincerely, 

Alexandra Tosun, PhD 

Associate Editor 

PLOS Medicine